# Effect of Rinse Solutions on *Rhizostoma pulmo* (Cnidaria: Scyphozoa) Stings and the Ineffective Role of Vinegar in Scyphozoan Jellyfish Species

**DOI:** 10.3390/ijerph20032344

**Published:** 2023-01-28

**Authors:** Ainara Ballesteros, Macarena Marambio, Carles Trullas, Eric Jourdan, Jose Tena-Medialdea, Josep-Maria Gili

**Affiliations:** 1Department of Marine Biology and Oceanography, ICM-CSIC-Institute of Marine Sciences, Passeig Marítim de la Barceloneta 37-49, 08003 Barcelona, Spain; 2ISDIN, Innovation and Development, C. Provençals 33, 08019 Barcelona, Spain; 3IMEDMAR-UCV-Institute of Environment and Marine Science Research, Universidad Católica de Valencia SVM, C. Explanada del Puerto S/n, 03710 Calp, Spain

**Keywords:** cnidarian, cnidocyst, cnidocyte, first-aid protocol, nematocyst discharge, pain, Scyphozoa class, venom

## Abstract

*Rhizostoma pulmo* is a widely distributed scyphozoan in the Mediterranean Sea. Their stings result mainly in erythema, small vesicles, or/and pain, and cause a high number of bathers to seek assistance from first-aid services during the summer season. Despite the threat that jellyfish stings represent to public health, there is disagreement in the scientific community on first-aid protocols, with the dispute largely centered around the effectiveness of vinegar. In the present research, we investigated the effect of commonly used rinse solutions on nematocyst discharge in *R. pulmo* and the effect of vinegar on three more scyphozoans (*Aurelia* sp., *Cassiopea* sp., and *Rhizostoma luteum*). Scented ammonia, vinegar, and acetic acid triggered nematocyst discharge in *R. pulmo*. Vinegar also caused nematocyst discharge in *Aurelia* sp., *Cassiopea* sp., and *R. luteum*. In contrast, seawater, baking soda, freshwater, urine, and hydrogen peroxide were considered neutral solutions that did not induce nematocyst discharge. These results indicate that the use of vinegar, acetic acid, or commercial products based on these compounds is counterproductive. Their use can worsen pain and discomfort caused not only by *R. pulmo* stings but also by those of any scyphozoan. The use of seawater is recommended for cleaning the *R. pulmo* sting site until an inhibitor solution that irreversibly prevents nematocyst discharge is discovered.

## 1. Introduction

Around 150 million people worldwide are exposed annually to jellyfish [1], representing a serious public health problem in some places [2]. The Cnidaria phylum, which includes jellyfish, is considered the oldest lineage of venomous animals [3]. Their toxicity is associated with the presence of cells unique to the phylum and known as cnidocytes [4,5]. The accidental encounter between jellyfish and humans triggers cnidocyst activation, a subcellular-enclosed capsule within cnidocytes with a mixture of toxins [5], causing stings in users of the marine environment [2,6,7,8]. Although encounters with animals belonging to the Cubozoa and Hydrozoa classes result in more serious envenomation [9,10], systemic effects and potentially fatal reactions can occur in the Scyphozoa class (“true jellyfish”) [9]. Sting severity varies across scyphozoan jellyfish species [11], but stings are generally characterized by low to moderate pain and cutaneous lesions [10,11]. Despite scyphozoan stings rarely requiring hospitalization [11,12,13,14], they often require first aid from lifeguard services [2,7,14,15,16] and, in some cases, medical attention [2,14,17,18].

After a cnidarian sting, there is a set of actions that can help to lessen the sting, with two clear objectives among the scientific community: (1) prevention of further cnidocyst discharge by cleaning the sting area with a rinse solution (e.g., seawater, vinegar, ammonia), and (2) limiting the action of the venom in terms of pain and tissue damage (e.g., heat, cold) [10]. While the goals of the guidelines seem to have consensus, disputes are mainly centered around the effectiveness of rinse solutions [9,10,19]. 

Vinegar is a well-known folk remedy employed as a rinse solution on the skin for cleaning off residual cnidocytes, but this recommendation is largely based on empirical knowledge [9,20]. While some research groups support the use of vinegar for cubozoans, hydrozoans, and scyphozoans [21], research aimed at clarifying its use in scyphozoan first-aid protocols warns against and contraindicates its use [22,23,24,25]. Although efficacy trials of vinegar or acetic acid for scyphozoans showed nematocyst discharge in many species [22,23,24,25], the most recent reviews on the topic called for further research [9,10].

In the Mediterranean region, the scyphozoan *Rhizostoma pulmo* is widely distributed throughout the basin and is the most common species in the western half [26]. Reports of blooms of this species have been frequent, even reporting very high abundances in some areas in recent years [26,27,28], with important socio-economic effects on some human activities such as tourism due to beach closures and human stings. In Spain, and specifically on the Catalonian coast, both *R. pulmo* and *Pelagia noctiluca* are the main jellyfish responsible for stings requiring first aid from lifeguard services during the bathing season, reaching thousands of incidents per season [16,29]. Collectively, their four different types of nematocysts (heterotrichous microbasic euryteles, holotrichous isorhizas, atrichous a-isorhizas, and atrichous α-isorhizas) produce a medium-severity sting involving local reactions such as erythema, small blisters, or/and pain [30,31,32]. Despite their high frequency in the Mediterranean Sea [26,27] and the high incidence of stings in bathers on some beaches, even exceeding the number seeking first aid for *P. noctiluca* stings [29], no research has been carried out on the efficacy of the rinsing solutions used in the first-aid protocol after their sting.

In order to avoid the extrapolation of results from other jellyfish species, our research aimed to achieve two clear objectives: (1) elaborate a species-specific first-aid protocol for *R. pulmo*; and (2) increase knowledge about the efficacy and role of vinegar in the nematocyst discharge in *Aurelia* sp. (mild-severity), *Cassiopea* sp. (mild-severity), and *Rhizostoma luteum* (medium-severity) [32]. These last three species were chosen because they are species belonging to the Scyphozoa class [32] and for which the effect of vinegar on the discharge of nematocysts is not known [32]. In this way, it is intended to know the effect of vinegar on species not previously evaluated, increasing the number of Scyphomedusae evaluated. This study contributes to the field of jellyfish sting management as it is the first study that evaluates the response of *R. pulmo* stinging cells to the different commonly used rinse solutions as well as the effect of vinegar on the nematocyst discharge from *Aurelia* sp., *Cassiopea* sp., and *R. luteum*. In addition, this research contributes to knowing the efficacy and role of vinegar in the first-aid protocols after scyphozoan jellyfish stings.

## 2. Materials and Methods

### 2.1. First-Aid Protocol Experiments

#### 2.1.1. Jellyfish Origin

Newly released ephyrae from *R. pulmo, Cassiopea* sp., and *Aurelia* sp. polyps were obtained through the natural strobilation process at the Institute of Marine Sciences (ICM-CSIC, Barcelona, Spain) and the Institute of Environment and Marine Science Research (IMEDMAR-UCV, Calpe, Spain). Ephyrae were grown in captive conditions to reach a juvenile stage. Individuals were fed ad libitum every day with *Brachionus* sp. (rotifers) and enriched *Artemia* sp. nauplii. *R. luteum* were collected from the beaches of Malaga (Spain) in August 2021 by lifeguard staff.

#### 2.1.2. Tentacle Solution Assay (TSA)—Nematocyst Discharge

The nematocyst response for *R. pulmo* was evaluated in the presence of eight rinse solutions: seawater (used as a control, locally collected and 10 μm filtered), vinegar (white vinegar, labeled: 6% acetic acid, Vivó Brand, Vinagres Parras S.A, Toledo (Spain)), 5% acetic acid (CAS Number: 64-19-7, Sigma-Aldrich Company, Merck Group, Darmstadt, Germany) in freshwater (tap water), scented ammonia (Bosque Verde Brand, The SPB Global Corporation S.L, Valencia, Spain), 10% baking soda (Hacendado Brand, Mercadona S.A, Barcelona, Spain) in seawater (locally collected and 10 μm filtered), freshwater (tap water), urine (freshly collected from a volunteer, pH = 5.80), and hydrogen peroxide (3% *w*/*v* hydrogen peroxide, Montplet Brand, Laboratorios Montplet S.A.U, Barcelona, Spain). In the case of *Aurelia* sp., *Cassiopea* sp., and *R. luteum*, only seawater (used as control, locally collected and 10 μm filtered) and vinegar (white vinegar, labeled: 6% acetic acid, Vivó Brand, Vinagres Parras S.A., Toledo, Spain) were used.

Due to the morphological variety among the Scyphozoa class, the study areas differed between species: the exumbrella for *R. pulmo* and *Cassiopea* sp. (Order: Rhizostomeae), the end of the appendages for *R. luteum* (Order: Rhizostomeae), and the marginal tentacles for *Aurelia* sp. (Order: Semaeostomeae).

Jellyfish pieces from several individuals (*R. pulmo* = 40 individuals, *Aurelia* sp. = 5 individuals, *Cassiopea* sp. = 10 individuals, and *R. luteum* = 3 individuals) were transferred to slides (76 × 26 mm). After the experiments had begun, pieces were observed under a light microscope to ensure the integrity of the nematocyst batteries. Afterward, 15 μL of each solution were applied to determine the effect on nematocyst discharge for a period of 30 s. 

The nematocyst response was classified qualitatively into four categories according to the discharge scale of Pyo et al. (2016) [22]:0: no discharge observed;+: low discharge of nematocysts;++: medium discharge of nematocysts;+++: maximum discharge of nematocysts.

Then, the effect of the rinse solution was classified into one of the following two categories: Activator effect solution: nematocysts were activated after the application of the solution;Neutral effect solution: nematocysts were not activated after the application of the solution.

## 3. Results

### 3.1. Response of Rhizostoma pulmo Nematocysts to Rinse Solutions

In order to determine the efficacy of eight commonly used rinse solutions (seawater, vinegar, 5% acetic acid in freshwater, scented ammonia, 10% baking soda mixed in seawater, freshwater, urine, and hydrogen peroxide), nematocyst discharge was evaluated after their application (Table 1). 

Seawater, 10% baking soda mixed in seawater, freshwater, urine, and hydrogen peroxide did not trigger nematocyst discharge in *R. pulmo* (Figure 1A,E–H), thus they were classified as neutral rinse solutions (Table 1). Rinse solutions of vinegar and 5% acetic acid produced medium discharge, and scented ammonia triggered immediate and massive discharge (Table 1). (Figure 1B–D, red arrows). Therefore, they were considered activator rinse solutions (Table 1).

### 3.2. Response of Aurelia sp., Cassiopea sp., and Rhizostoma luteum Nematocysts to Vinegar

The effect of vinegar, the compound that generates the most debate and controversy in first-aid protocols, was tested in three other scyphozoan species (Table 2). Seawater was used as a control (Table 2).

Vinegar triggered rapid nematocyst discharge in all the scyphozoans (Figure 2D–F, red arrows), and so it was classified as an activator rinse solution (Table 2). However, seawater did not induce nematocyst discharge in *Aurelia* sp., *Cassiopea* sp., and *R. luteum* (Figure 2A–C). It was considered a neutral rinse solution (Table 2).

## 4. Discussion

Scyphozoan jellyfish stings are a human health hazard in the Mediterranean Sea [2,7,14,16]. Among them, *R. pulmo* stings have a high incidence in beach-goers [14,16] due to their wide distribution in the Mediterranean basin [26] and have a medium sting severity [32]. Despite the threat that jellyfish stings represent, systematic reviews have shown a lack of knowledge about the efficacy of rinse solutions used after scyphozoan jellyfish stings in general [10] and *R. pulmo* in particular [10,11,13,19]. For the first time, we evaluated the efficacy of different rinse solutions following a *R. pulmo* sting. Vinegar, 5% acetic acid, and scented ammonia were identified as activating solutions, while seawater, 10% baking soda mixed in seawater, freshwater, urine, and hydrogen peroxide were classified as neutral solutions that did not trigger nematocyst discharge (Table 1). 

Vinegar or acetic acid solutions are rinse solutions commonly recommended for cleaning the sting area after a cnidarian sting [10,21,33]. In the present study, immediate nematocyst discharge was observed after the application of vinegar and acetic acid to *R. pulmo* (Figure 1B,C and Table 1). Thus, we conclude that the use of these rinse solutions is detrimental to the removal of tissue and residual cnidocytes after an *R. pulmo* sting.

The most recent review on jellyfish sting first-aid assistance and treatments suggested further research to clarify the role of vinegar in scyphozoan first-aid guidelines [9,10]. In the present study, in addition to *R. pulmo*, the nematocysts of *Aurelia* sp., *Cassiopea* sp., and *R. luteum* were also activated after the application of vinegar (Figure 2D–F and Table 2). In previous studies, vinegar or acetic acid promoted discharge in scyphomedusae such as *Chrysaora quinquecirrha* [25], *Cyanea capillata* [24,34], *Nemopilema nomurai* [22], and *P. noctiluca* [23,35]. Overall, the scientific evidence contraindicates rinsing the sting area with vinegar, acetic acid, or commercial products based on these compounds for the Scyphozoa class. Our results demonstrate that its use is detrimental in scyphozoan first-aid protocols, in contrast to its wide and consensual acceptance to prevent the discharge of unfired nematocysts of cubozoans [22,23,33,36]. Class-specific, or even species-specific, first-aid protocols should be considered when making recommendations on the first-aid protocols for cnidarian stings, with special emphasis on this contraindication for the Scyphozoa class.

Urine, urea, or ammonia are well-known folk remedies for treating jellyfish stings [10,11]. On the Italian coast, 59.3% of non-pharmacological treatment consisted of washing the sting area with ammonia [14]. Conversely, ammonia has been recognized as a solution that causes cnidocyst discharge in Scyphozoa, Hydrozoa, Cubozoa, and Anthozoa [8,25,37]. In the present research, scented ammonia was also considered an activator solution for *R. pulmo* (Figure 1D and Table 1), in agreement with the findings of Birsa et al. (2010) [25] for *C. quinquercirrha*. In contrast, unfired nematocysts were observed after the application of urine (Figure 1G), as reported in previous research [25,35]. Our results are in contrast to those of Doyle et al. (2017) [21] in *C. capillata*; therefore, in order to clarify the role of urine, further research with other scyphozoan species is needed.

Rinsing with deionized, distilled, or freshwater has been contraindicated due to the induction of discharge by osmotic pressure [38]. Nevertheless, methods that have directly evaluated the effect on cnidocyst discharge (e.g., Tentacle Solution Assay [33]) have shown no solid evidence to support this contraindication [8,22,25,33,35,39]. Pyo et al. (2016) [22] did not observe discharge after the application of distilled water in both the cubozoan *Carybdea mora* and the scyphozoan *N. nomurai*. Wilcox et al. (2017) [39] did not detect a significant discharge for the hydrozoan *Physalia utriculus*, and Ballesteros et al. (2022) [8] observed a low discharge for the anthozoan *Anemonia viridis*. Despite not inducing discharge (Figure 1F and Table 1), deionized, distilled, and freshwater are widely used in scientific experiments to isolate intact cnidocytes from the ectoderm of scyphozoans [40,41]. In any case, from the perspective of first-aid protocols, their use as rinsing solutions could imply the potential detachment of residual tissue and subsequent cnidocyst activation by mechanical effect. Hence, deionized, distilled, or freshwater are not recommended.

Compounds such as seawater, baking soda, and hydrogen peroxide have also been tested previously [11]; in the present study, none of them produced discharge in *R. pulmo* (Table 1). Baking soda promoted cnidocyst discharge in species of the Scyphozoa, Cubozoa, Hydrozoa, and Anthozoa classes [8,36,39,42]. As a precautionary measure, the use of baking soda was removed from a recent version of the Spanish Mediterranean first-aid guidelines [32] until its role in the Scyphozoa class is clarified.

Seawater is a neutral solution in cnidarians (Table 1 and Table 2) [8,21,22,23,33,39]. However, discrepancies exist among the scientific community about its effectiveness as a rinse solution. Since seawater is not an inhibitor solution, researchers warn of cnidocyst activation if the rinsing process causes them to roll on human skin, allowing inoculation with a second venom load [21,23]. The area affected by a cnidarian sting must, above all, be cleaned with an inhibitor solution to irreversibly prevent the discharge that triggers a second envenomation. Although seawater is not an inhibitor solution, having excluded the activating solutions per se (Table 1), seawater is recommended for cleaning the sting area of *R. pulmo* as it is an easily accessible solution for rescue services and marine environment users. Due to the paucity of studies in Scyphomedusae, knowledge on solutions that irreversibly prevent the discharge of *R. pulmo* nematocysts is urgently needed to propose a safe framework of action for stakeholders (e.g., beach-goers, lifeguards, and a centralized government) for the first-aid assistance of these stings. To carry out the first steps of complete assistance, this research recommends: (1) removing residual tissue with a rigid element (e.g., tweezers or a credit card) and (2) cleaning the sting area with seawater. The use of vinegar as a rinse solution is contraindicated because it is an activator solution. After cleaning the sting area with seawater, the guidelines suggest (3) applying ice packs for 15 min in total (3 min of application + 2 min of rest (×3)), and (4) if the pain persists, going to the nearest health center for medical assistance [21,23].

## 5. Conclusions

*Rhizostoma pulmo* blooms are frequent in some coastal areas; this is the main jellyfish responsible for stings requiring first aid from lifeguard services during the bathing season. Despite the high incidence of stings in bathers, no research has been carried out on the efficacy of the rinsing solutions used in the first-aid protocol after their sting. Our research shows how vinegar, 5% acetic acid, and scented ammonia produce nematocyst discharge, and they are not recommended for cleaning the sting site after *R. pulmo* stings. On the contrary, 10% baking soda mixed in seawater, freshwater, urine, and hydrogen peroxide does not produce nematocyst discharge after its application. Of all of them, seawater is recommended as it is an easily accessible solution for rescue services and marine environment users. However, this research highlights the need to find an inhibitory rinse solution to replace seawater (a neutral solution).

Finally, the use of vinegar is not recommended for cleaning the sting area of scyphozoan jellyfish stings, as its application produces nematocyst discharge, not only in *R. pulmo* but also in *Aurelia* sp., *Cassiopea* sp., and *Rhizostoma luteum*. The sting of these jellyfish species could worsen with the use of an activator rinse solution that increases the venom load during the cleaning step in first-aid protocols.

## Figures and Tables

**Figure 1 ijerph-20-02344-f001:**
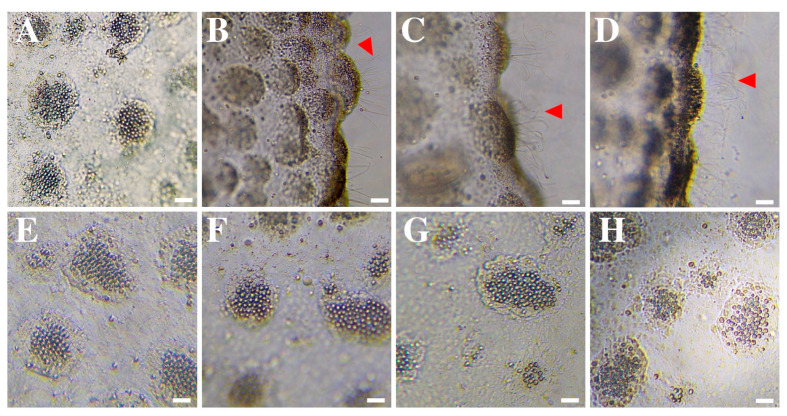
Nematocyst response of *Rhizostoma pulmo* after the application of (**A**) seawater (control), (**B**) vinegar, (**C**) 5% acetic acid, (**D**) scented ammonia, (**E**) 10% baking soda mixed in seawater, (**F**) freshwater, (**G**) urine, and (**H**) hydrogen peroxide. Note that nematocyst discharge (red arrows) occurred with (**B**) vinegar, (**C**) 5% acetic acid, and (**D**) scented ammonia. Scale bars: 50 μm.

**Figure 2 ijerph-20-02344-f002:**
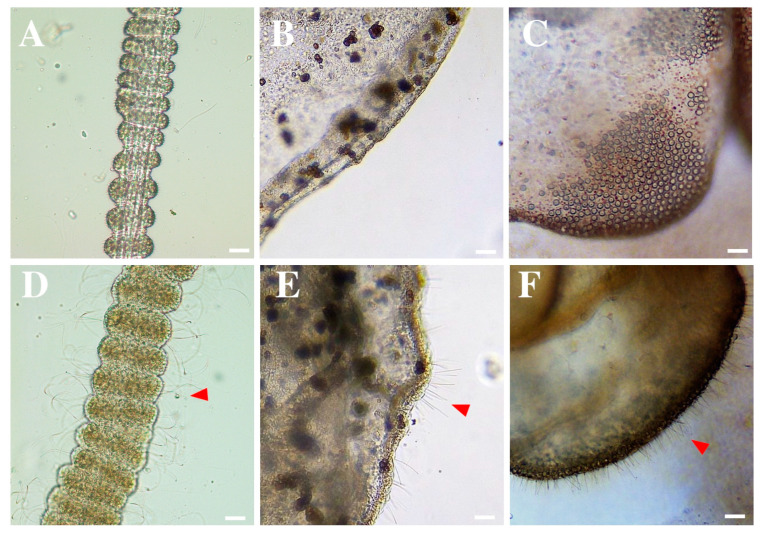
Nematocyst response in *Aurelia* sp. (**A**,**D**), *Cassiopea* sp. (**B**,**E**), and *Rhizostoma luteum* (**C**,**F**) after the application of (**A**–**C**) seawater (control) and (**D**–**F**) vinegar. Note the nematocyst discharge (red arrows) after vinegar application. Scale bars: 50 μm.

**Table 1 ijerph-20-02344-t001:** *Rhizostoma pulmo* nematocyst response after the application of different rinse solutions commonly recommended in first-aid protocols. The number of replicates was 10 for each rinse solution.

Rinse Solution	Discharge ^1^	Effect ^2^
Seawater (control)	0	Neutral
Vinegar	++	Activator
5% acetic acid in freshwater	++	Activator
Scented ammonia	+++	Activator
10% baking soda mixed in seawater	0	Neutral
Freshwater	0	Neutral
Urine	0	Neutral
Hydrogen peroxide	0	Neutral

^1^ Nematocyst discharge categories: 0 = no discharge was observed; ++ = medium discharge of nematocysts; +++ = high discharge of nematocysts. ^2^ Rinse solution categories: neutral solution = nematocysts are not activated after the application of the solution; activator solution = nematocysts are activated after the application of the solution.

**Table 2 ijerph-20-02344-t002:** Nematocyst response in scyphozoan jellyfish after the application of seawater (control) and vinegar. The number of replicates was ten for each rinse solution in *Aurelia* sp. and *Cassiopea* sp., and three in *Rhizostoma luteum*.

***Aurelia* sp.**		
**Rinse Solution**	**Discharge ^1^**	**Effect ^2^**
Seawater (control)	0	Neutral
Vinegar	++	Activator
***Cassiopea* sp.**		
**Rinse solution**	**Discharge ^1^**	**Effect ^2^**
Seawater (control)	0	Neutral
Vinegar	++	Activator
** *Rhizostoma luteum* **		
**Rinse solution**	**Discharge ^1^**	**Effect ^2^**
Seawater (control)	0	Neutral
Vinegar	++	Activator

^1^ Nematocyst discharge categories: 0 = no discharge was observed; ++ = medium discharge of nematocysts. ^2^ Rinse solution categories: neutral solution = nematocysts are not activated after the application of the solution; activator solution = nematocysts are activated after the application of the solution.

## Data Availability

Data sharing is not applicable. In this study, no new data were created or analyzed.

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
