# Peer review of "Effect of Rinse Solutions on *Rhizostoma pulmo* (Cnidaria: Scyphozoa) Stings and the Ineffective Role of Vinegar in Scyphozoan Jellyfish Species"

_ijerph, 2023, doi:10.3390/ijerph20032344_

Round 1

Reviewer 1 Report

Reviewer 1, states that the MS submitted by Ballesteros et. al. meets the scientific standards and recommends the paper for publication in its revised form.

Author Response

Thank you very much for your comments.

Reviewer 2 Report

I provided significant comments in my previous reviews and I feel the authors addressed all my concerns with this resubmission. I apologize for the delay in my review/comments, I received a request to review during a busy holiday season followed by a research conference. This is great piece I plan to use in my future classes of introduction to marine biology because I think it'd be of wide interest to students. I feel like this fits well as a communication article within the journal.

Author Response

If this report corresponds to the previous ones, we apply all the changes and suggestions of the previous reviewers. Thank you for your comments and improvements.

Reviewer 3 Report

This manuscript clearly and concisely describes the effect of commonly used rinse solutions on nematocyst discharge in four species of scyphozoans and might be published in the journal of Environ. Res. Public Health after some revisions.

Major comments:

P2 L76

It is not clear, why  Aurelia sp., Cassiopea sp., and R. luteum species was chosen for this investigation. There is no information about these species in the introduction before line 76.

P2 L78-79

The statment " the first study that evaluates the response of R. pulmo stinging cells to the different commonly-used rinse solutions" gives the impression that there is information about the other three mentioned species (Aurelia sp., Cassiopea sp., R.  luteum) which was not included in the introduction.

What purpose were two similar solutions (6%  vinegar and 5% acetic acid) used  for?

It is not clear, what is scented ammonia.  Is it liquid, was it dissolved or diluted in water (sea or tap water?). Is it safe to apply the scented ammonia to the skin?

P3 L98

The concentration of  the hydrogen peroxide must be specified.

P4 L552

It should  be explained in the section 3.2., why only vinegar effect  was tested on Aurelia sp., Cassiopea sp. and Rhizostoma luteum nematocysts.

P3 L127, L132

Are all of the listed rinse solutions  really commonly used? Please, add the reference to prove that solutions  are "commonly recommended in first-aid protocols".

4. Discussion

It would be beneficial to discuss the possible compounds and mechanism of inhibitory solution action, current progress  in the development of this solution. Same for the role of factors determined the applicability of a solution for the safe cnidocyst removal (pH close to that of sea water, osmotic pressure, denaturing effect of urea). It is logical to assume that if pH is very different from the pH of sea water, it can lead to the cnidocyst discharge. In this regard, the pH of the solutions used could be indicated and shown in the tables or in the material and method section.

Minor revisions:

P1 L35-37

I suggest to divide the sentence "The Cnidaria phylum, which 35 includes jellyfish, is considered the oldest lineage of venomous animals [3], and their toxicity is associated with the presence of cells unique to the phylum and known as cnidocytes [4,5]. "  into two sentences.

P1 Line 40-41

I suggest to change "species" to "animals" or "organisms"

P2 L80

"clarifies " recommended to replace to "contributes to"

P2 L67-68

In the sentence the mention of  "four different types of nematocysts" draws attention, while the authors do not further mention the existence of different types of nematocysts. This raises questions, but the main idea of the sentence, I think, is the consequences of sting. Please, specify the information about nematocyst types or delete it.

Author Response

Please see the attachment, 

Thank you very much for your comments.

Reviewer 4 Report

Introduction:

complement with the review work: https://onlinelibrary.wiley.com/doi/10.1111/ijd.15804

Material and methods: 

include meaning at first mention ICM-CSIC and IMEDMAR-UCV

include the pH of the urine, since depending on its pH it may or may not activate the discharge of nematocysts.

Results:

add an image of each of the jellyfish species considered in the study.

Discussion:

Do not generalize the use of urine and consider pH of urine, review other studies, for example, https://www.mdpi.com/2072-6651/9/7/215

https://www.ncbi.nlm.nih.gov/pmc/articles/PMC4962017/

Author Response

Thanks for your comments.

Round 2

Reviewer 3 Report

This manuscript clearly and concisely describes the effect of commonly used rinse solutions on nematocyst discharge in four species of scyphozoans and might be published in the International Journal of Environmental Research and Public Health journal in present form.